# Optimizing Mandibular Advancement Maneuvers during Sleep Endoscopy with a Titratable Positioner: DISE-SAM Protocol

**DOI:** 10.3390/jcm11030658

**Published:** 2022-01-27

**Authors:** Patricia Fernández-Sanjuán, Juan José Arrieta, Jaime Sanabria, Marta Alcaraz, Gabriela Bosco, Nuria Pérez-Martín, Adriana Pérez, Marina Carrasco-Llatas, Isabel Moreno-Hay, Marcos Ríos-Lago, Rodolfo Lugo, Carlos O’Connor-Reina, Peter Baptista, Guillermo Plaza

**Affiliations:** 1Department of Maxillofacial Surgery and Dentistry, Hospital Universitario San Francisco de Asís, Universidad Rey Juan Carlos, 28002 Madrid, Spain; patriciafsanjuan@yahoo.es; 2Dental Sleep Medicine, Hospital Universitario Sanitas La Zarzuela, 28942 Madrid, Spain; 3Department of Stomatology, Hospital Universitario Fundación Jiménez Díaz, 28040 Madrid, Spain; jjarrietab@yahoo.es; 4Department of Otolaryngology Head and Neck Surgery, Hospital Universitario Fundación Jiménez Díaz, 28040 Madrid, Spain; JSanabria@fjd.es; 5Department of Otolaryngology Head and Neck Surgery, Hospital Universitario Sanitas La Zarzuela, 28942 Madrid, Spain; malfue@hotmail.com (M.A.); maria.bosco@salud.madrid.org (G.B.); n.perezmartin@hotmail.com (N.P.-M.); 6Department of Otolaryngology Head and Neck Surgery, Hospital Universitario de Fuenlabrada, Universidad Rey Juan Carlos, 28942 Madrid, Spain; 7Department of Otolaryngology Head and Neck Surgery, Hospital La Milagrosa, 28010 Madrid, Spain; madrianaperez2004@yahoo.com; 8Department of Otolaryngology Head and Neck Surgery, Hospital Universitario Dr. Peset., 46017 Valencia, Spain; marinacll@gmail.com; 9Division of Orofacial Pain, Department of Oral Health Science, College of Dentistry, University of Kentucky, Lexington, KY 40536, USA; imo226@uky.edu; 10Department of Basic Psychology II, Faculty of Psychology, UNED, 28040 Madrid, Spain; mrios@psi.uned.es; 11Department of Otolaryngology Head and Neck Surgery, Hospital San José, Monterrey 64718, Mexico; rodo_lugo@me.com; 12Department of Otolaryngology Head and Neck Surgery, Hospital Quironsalud Marbella, 29603 Marbella, Spain; coconnor@us.es; 13Department of Otolaryngology Head and Neck Surgery, Hospital Quironsalud Campo de Gibraltar, 11379 Palmones, Spain; 14Department of Otolaryngology Head and Neck Surgery, Clínica Universitaria de Navarra, 31008 Pamplona, Spain; peterbaptista@gmail.com

**Keywords:** sleep breathing disorders, mandibular advancement, MAD, titratable positioner, drug-induced sleep endoscopy, patient selection

## Abstract

Mandibular advancement devices (MAD) are an effective alternative treatment to CPAP. Different maneuvers were performed during drug sleep-induced endoscopy (DISE) to mimic the effect of MAD. Using the Selector Avance Mandibular (SAM) device, we aimed to identify MAD candidates during DISE using a titratable, reproducible, and measurable maneuver. This DISE-SAM protocol may help to find the relationship between the severity of the respiratory disorder and the degree of response and determine the advancement required to improve the collapsibility of the upper airway. Explorations were performed in 161 patients (132 males; 29 females) with a mean age of 46.81 (SD = 11.42) years, BMI of 27.90 (SD = 4.19) kg/m^2^, and a mean AHI of 26.51 (SD = 21.23). The results showed no relationship between severity and MAD recommendation. Furthermore, there was a weak positive relationship between the advancement required to obtain a response and the disease severity. Using the DISE-SAM protocol, the response and the range of mandibular protrusion were assessed, avoiding the interexaminer bias of the jaw thrust maneuver. We suggest prescribing MAD as a single, alternative, or multiple treatment approaches following the SAM recommendations in a personalized design.

## 1. Introduction

Obstructive sleep apnea (OSA) is a highly prevalent disease with significant public health outcomes [1] whose primary symptom is excessive daytime sleepiness that, together with mood alteration and cognitive impairment, produces a progressive deterioration in the quality of life of patients. In addition, it has been associated with an increased risk of arterial hypertension, cardiovascular morbidity and mortality, and occupational and traffic accidents [2,3].

Although continuous positive air pressure (CPAP) is considered the gold standard treatment for OSA, it is estimated that CPAP failure occurs in approximately 35% of patients [4] and that 29–83% of users are nonadherent to this therapy [5]. Mandibular advancement devices (MAD) are an effective alternative treatment to CPAP for many of these patients. Their mechanism of action is based on the advancement and stabilization of the mandible and the hyoid. They increase basal electromyographic activity of the genioglossus, tighten the palatoglossus and palatopharyngeus muscles, increase the vertical dimension, and activate the lateral walls of the velopharynx [6,7,8,9,10]. Furthermore, the soft palate is displaced ventrally, increasing the retromandibular space’s caliber [6,7]. MAD are indicated in patients with simple snoring, mild and moderate OSA without comorbidity and severe OSA patients who refuse or do not tolerate CPAP [11]. Although CPAP is more effective in reducing the apnea–hypopnea index (AHI), MAD is preferred by patients and their partners as they are more portable, make no noise, are more comfortable, and are better tolerated [12,13]. However, individual variability exists in response to MAD treatment [14]. Some studies show that MAD are effective in 30–81% of cases [15]. This high variability in response suggests the need for the identification of potential responders. 

There are numerous studies aimed at identifying suitable candidates for treatment with MAD on occasions with contradictory results. Despite it is accepted that individuals with less severe and predominantly supine OSA respond better [16,17], some publications find MAD to also be a good alternative for severe patients [18] and non-positional OSA [14]. Considering anthropometric variables, younger patients with low body mass index or female candidates may also have better responses [19,20,21,22]. Cephalometric variables are considered inconsistent predictors, and therefore reliance on them for this purpose is discouraged [23]. Some values for air pressure have even been identified as having very high predictive values in detecting MAD non-responder patients [24,25,26]. Regarding polysomnographic variables, some studies agree that MAD therapy is more effective in non-REM related OSA [14]. Considering the multifactorial origin of OSA [27], a more significant response to oral therapy occurred in patients with a mild anatomical compromise and a lower loop gain [28,29]. Finally, Sutherland et al. [30] performed a study on awake multimodal phenotyping to predict the MAD treatment outcome. They found that it does not enhance outcome prediction, concluding that office-based awake assessments have limited utility for robust clinical prediction models and suggested that future work should focus on sleep-related assessments. However, due to the growing interest in MAD as a therapeutic option and knowing that OSA is not adequately alleviated in all patients wearing MAD, it is mandatory to find an objective and reproducible procedure to help with patient selection.

To date, drug-induced sleep endoscopy (DISE) provides the most helpful information on upper airway (UA) collapsibility during sleep compared to other evaluation techniques available [31,32]. DISE was first described by Croft and Pringle [33] and has become widely used. It involves the assessment of the UA by means of fiberoptic endoscopy under pharmacologic sedation, which simulates sleep. Nowadays, this procedure is performed in patients seeking alternatives to CPAP treatment. To mimic the effect of MAD, different maneuvers have been used. For example, it has been shown that the results obtained by manually performing jaw thrust maneuvers do not always correlate with those achieved by intraoral appliances [34,35,36,37]. The possible cause of this difficulty could be inconsistency in the application of the maneuver, inaccurate measurement of the advancement in some cases, or patient stimulation that could lead to arousal [38]. 

The aim of the present study is twofold. First, to identify those patients during DISE who will potentially benefit from oral appliance therapy using a titratable, reproducible, and measurable maneuver with the Selector Avance Mandibular (SAM) device. Second, to study the relationship between the severity of the respiratory disorder and the degree of response and the degree of advancement required to respond during DISE. In addition, it may also suggest different levels of recommendation to optimize MAD treatment in a personalized design according to the expected response. 

## 2. Materials and Methods

### 2.1. Participants

A prospective study was designed. The Medical Ethics Committee of Hospital Universitario de Fuenlabrada approved the study protocol in accordance with the Declaration of Helsinki (APR 20/30). Data on study subjects were collected and stored anonymously to protect personal information.

The study included one hundred and sixty-one patients (132 males; 29 females). The inclusion criteria were adults (>18 years) with snoring and OSA. The patients were evaluated using the standard full-night polysomnography and interpreted using the diagnostic criteria established by the American Academy of Sleep Medicine [39]

The exclusion criteria were propofol allergy, periodontal disease, insufficient number of teeth, acute or severe temporomandibular disorder, or an exaggerated gag reflex. Patients with medical history of congenital abnormalities of the UA were also excluded. 

### 2.2. Methods

The study protocol included complete medical history with the following variables: age, sex, body mass index (BMI), and Epworth Sleepiness Scale (ESS) score. All the patients underwent diagnostic sleep polysomnography (PSG). All the patients underwent an office examination including awake UA endoscopy examination to identify anatomical alterations. Orthopantomography, periodontal, dental, and muscular/articular examinations were also performed in the dental office to confirm each patient’s suitability to use MAD.

Subsequently, the anesthetic assessment was performed prior to DISE. The procedure was performed in an operating theatre as a day-case technique. The clinicians involved in the procedure included an anesthesiologist, an otolaryngologist, and a dentist. A flexible video endoscope (TGH-Endoscopia, MACHIDA ENT-30PIII, Madrid, Spain) was used to visualize the UA. The classification system used to evaluate the UA patterns and areas of collapse was the VOTE scale [40]. 

The DISE-SAM protocol was then started. To mimic a mandibular advancement splint therapy, it incorporates a device called Selector Avance Mandibular (SAM) that was designed to produce personalized mandibular advancement during DISE by the first author. The SAM is a titratable (40 g) tool that allows the operator to perform controlled, progressive, and high-precision advancements (in the mm range) during sedation, starting from a neutral/retrusive position until reaching the optimal position in which, if the patient responds, apnea/snoring may be eliminated. The device has two parts: a body that is reusable and the disposable trays to be inserted into the patient´s mouth. The trays can be positioned or removed independently from the device body which facilitates insertion or removal from the mouth at any time (Figure 1 and Appendix A). 

Before starting the sedation, the SAM was personalized to the patient’s dental arches with polyvinylsiloxane dental impression material. First, when the patient was awake, he or she was asked to move the jaw freely forward and backward. The maximum retrusive and the maximum protrusive positions were annotated, limiting the mandibular movement, so as not to force any position beyond the tolerable movement during sedation. The operator manually caused protrusive and retrusive movements by turning a thumbwheel at his/her convenience, depending on the sleep test requirements. Using the SAM, the mandibular position can be changed as many times as required without removing the device from the patient’s mouth, choosing the mandibular position to be fixed in the place of interest. Following the European position paper recommendations on DISE [31], propofol was used in this study. A 2% propofol syringe targeted controlled infusion pump was used with a target concentration of 2 ng/mL with progressive increases of 0.2–0.5 ng/mL as required. The sedation level was monitored using the bispectral index (BIS) (BIS Quatro, Covidien ILc, Mansfield, MA, USA). Levels between 50 and 70 BIS should be maintained. No vasoconstrictors or anesthetics were used. The fiberscope was introduced and placed in the nasopharynx where the operator waited until the appearance and visualization of two cycles of snoring, apnea, and breathing. Then, the UA was explored up to the larynx, analyzing the behavior at different VOTE levels. The neck and/or trunk was moved to lateral decubitus position to assess positional therapy [41]. The locations of the obstructions, the degree and pattern of collapse were carefully noted.

Once the UA was studied in the patient’s baseline situation, the SAM with the trays customized to the dental arches was introduced in the patient’s mouth. Although the device allows analyzing the UA at any position within the tolerable range of mandibular protrusive positions, the examination is performed in 0%, 50%, 75%, and 100% advancements (Figure 1). No response is expected at 0% advancement given that this is the maximum retrusive position. However, changes may occur in some patients due to reduction in the vertical dimension when the trays are fitted. Then, 50% advancement was executed. If none or insufficient response was detected at 50%, advancement to 75% was produced and, if there was no response, 100% advancement was finally carried out. When a good response was found, a slow decrease was performed until reaching the minimum effective advancement which might be at any point between the last two points tested. A valid response (50%, 60%, 70%, 75%, etc.) was considered when the patient remained at least 3 min without apnea and controlled O_2_ saturation. After this, lateral decubitus position was assessed with the MAD in the effective position. Thus, the assessments were performed in supine position and lateral head or trunk rotation with jaw advancement to evaluate the effect of a MAD, the effect of positional therapy, and to mimic combined treatment (positional therapy plus MAD) (Figure 1 and Figure 2).

Once the response to the jaw advancement was visualized, the SAM was withdrawn. The locations of the obstructions, the degree and pattern of collapse with the SAM were again noted to be compared with the baseline situation. This procedure usually takes less than 20 min. The first 10 min is for basal examination from the moment the patient acquires the adequate sedation and the last 10 min are invested in the advancement maneuvers and positional changes. Only one examination could not be completed because of sneezing, coughing, and excessive expectoration.

A scale of four levels of MAD recommendation was used, depending on the degree of response and the need for combined treatment. First, the S Recommendation (single treatment) is applied if all collapses are eliminated (VOTE 0000). Second, the A Recommendation (alternative treatment) is applied if the collapses are improved by at least 50%, changing from 2 to 1 or from 1 to 0, with some VOTE levels remaining at 1. If collapse 2 persists at one level, it must be momentary and should not be responsible for apnea, maintaining normal oxygen saturation. Snoring is expected due to persistent vibration and partial collapse at some level that could lead to residual apnea. A partial response is expected but results may improve with therapy combination. Third, the M Recommendation (multiple treatment approach) is applied when a collapsibility improvement is found, but a complete collapse responsible for apnea remains at any VOTE level despite the advancement maneuver. It is applied in combination with UA surgery or CPAP. Lastly, the No Recommendation is applied when a MAD does not change or worsens collapsibility. Likewise, it is also applied if maximum advancement is needed to achieve significant changes in UA patency since low patient tolerance is to be expected. These four levels are summarized in Table 1. 

## 3. Data Analyses

Kolmogorov–Smirnov normality tests were used to determine whether the distribution of continuous variables (age, AHI, BMI, and ESS) fitted the normal distribution in each group (*p* > 0.05 in all the cases). Thus, one-way ANOVA was applied to explore the existence of differences between the groups in age, BMI, and ESS (Bonferroni correction was applied for post hoc analyses). Variable “SAM recommendation” was constructed according to the criteria established in the procedure. In addition, a new variable MAD YES/NO was designed according to whether a MAD was recommended or not. Statistical analyses of categorical variables were performed using the chi-squared test. This was applied to study a relationship between severity and gender, “SAM recommendation” and “MAD YES/NO”. To explore the relationship between severity and advancement percentage, Somers’ D test was applied. It is an ordinal directional measure that indicates the strength and direction of the relationship between variables. A probability value less than 0.05 was considered to indicate statistical significance, and all the reported probability values were two-tailed. Finally, the SPSS v24.0 statistical software package was used to perform analyses.

## 4. Results

In total, 161 patients were included for analysis. The patients’ mean age was 46.82 ± 11.42 years; the mean BMI was 27.9 ± 4.19 kg/m^2^; the mean AHI was 26.43 ± 21.23 events per h; and the mean ESS was 9.87 ± 5.49. The participants were assigned to one of the four groups based on the AHI score as follows: snoring group, AHI = 0–4.9 events per h; mild OSA group, AHI = 5–14.9 events per h; moderate OSA group: AHI = 15–29.9 events per h; and severe OSA group, AHI > 30 events per h. The demographic characteristics of the four severity groups are summarized in Table 2.

All the groups were matched for age (F = 1.97; *p* = 0.12) and ESS (F = 1.31; *p* = 0.28) but not for BMI (F = 5.49; *p* = 0.001). Bonferroni’s post hoc comparison showed differences in BMI between the snoring group and the severe group, and between the mild group and the severe group (*p* = 0.005 and *p* = 0.007, respectively). No differences in gender were found between the groups (chi-squared test = 5.27; *p* = 0.15). 

MAD were not recommended in 26.1% of the patients. Of the 73.9% of the patients who did receive a recommendation for MAD therapy, 19.3% received the S recommendation, 34.2% received the A recommendation, and 20.5% received the M recommendation. Table 3 shows the percentage and number of patients classified according to severity who received the different SAM recommendations (see also Figure 3).

Furthermore, the percentages and the number of patients in each group of severity receiving different recommendations were calculated (Table 4 and Figure 4).

The results showed no relationship between severity and MAD YES/NO (chi-squared test = 3.75; *p* = 0.29) and SAM recommendation (chi-squared test = 13.94; *p* = 0.12) (see Figure 5).

The results indicate a positive relationship between severity and advancement percentage (d = 0.15; *p* = 0.048). Figure 6 shows the necessary mandibular advancement percentage to obtain response during DISE according to severity. There were patients in all the groups who responded with small advancements, and on the other hand, patients who required a larger protrusion to visualize improvement of collapsibility.

## 5. Discussion

To our knowledge, this is the first study evaluating a combination of DISE with a manually controlled titratable mandibular positioner, SAM, as a prediction tool for oral appliance treatment outcomes. This study arose due to the high interindividual variability in performing the jaw thrust maneuver during DISE, the great variability in response to MAD, and the need not only to identify responders, but also to propose different recommendations according to the expected degree of response (SAM recommendation).

Our results after the DISE-SAM protocol suggest that MAD therapy could be recommended to 73.9% of the patients, including S, A, and M recommendations (see Table 1). Our results expect total or partial response in 53.5% (S and A recommendation) of patients. For Sutherland et al. [14], 64% of the patients using MAD had total or partial responses in a meta-analysis. Metha et al. [13] found similar results (complete or partial response) in 62.5% of the patients. The differences in these percentages with our results may be caused by a higher proportion of moderate and severe OSA patients in our study. According to the proposed SAM recommendation, in 19.3% of the cases, MAD could be used as a single therapy (S recommendation) expecting a complete response. In 34.2% of the patients, it could be proposed as an alternative therapy (A recommendation), in which partial response was expected. Finally, for 20.5% of the patients (M recommendation), it could be recommended only in combination with selected therapies as it could provide a synergistic effect. MAD were not recommended in 26.1% of the patients. This value was slightly lower than the percentage of non-responders mentioned in Sutherland’s meta-analysis [14] or Metha’s study [13] who stated that 36% and 37.5% were non-responders. Again, this difference may be due to the incorporation of a new recommendation (M) for some patients who would benefit from multiple therapies for the first time considered. In other words, with this DISE-SAM protocol and proposing MAD YES/MAD NO, different levels of recommendation could be suggested, increasing the quality of the therapeutic recommendation.

In addition, different SAM recommendations showed no relationship with severity. According to our results, MAD could be proposed for severe OSA in 9.3% of patients as an S recommendation, in 25.9%—as an A recommendation, in 29.6%—as an M recommendation. In this view, 64.8% of severe OSA patients could benefit from MAD therapy. Perhaps, the recommendation based on a single variable, in this case, severity, is insufficient. Our observation suggests that the recommendation of MAD based on the severity of OSA is not accurate, as in all the severity groups, recommendations from S to No could be made (Figure 4).

Thus, the DISE-SAM protocol could also be useful to personalize each responder’s mandibular advancement regardless of severity. Our results found a positive relationship between the advancement required to obtain response and severity. However, this relationship was weak, given the low value for the statistical test. Finally, we found patients in all the severity groups with good responses to minor advancements, and others where maximal advancements were necessary to resolve the collapses (see Figure 6). MAD treatment could be not tolerable for these patients, with the necessary advancement being the largest, defining in advance that some patients would not be good candidates for this therapy. On the other hand, it has been reported that up to 39% of patients may respond with any advancement [42], and that in one third of patients, protruding the tongue or closing the mouth worsens the collapsibility [43]. Furthermore, some results show that a larger advancement is not always associated with a corresponding reduction in the AHI [44,45]. Knowledge of these aspects is very useful in treatment decision-making.

Likewise, there is no consensus on the starting protrusive position of the oral appliance (the position in which the MAD is manufactured and delivered for subsequent gradual adaptation) nor on the recommended therapeutic position. This is due to high individual variability between practitioners for the former [10,46,47] and high interindividual variability in response to the MAD therapy for the latter. Therefore, having information on the advancement required to alleviate the collapsibility would be helpful in planning the starting point of advancement and the target position to be reached.

Studies looking at the effect of mandibular advancement during DISE [48,49,50,51] performed manual jaw thrust maneuvers, with some authors [34,35,36,37] observing that the effect on the UA of a mandibular hyperprotrusion or chin lift maneuvers does not always correlate with the results obtained with a MAD. It has been demonstrated that jaw thrust shows a greater improvement at the hypopharyngeal level, but it is less effective in improving the obstruction at the retropalatal level than a MAD [37]. However, the same authors [52] also concluded that both a manual jaw thrust, and a temporary MAD seem to mimic the treatment with a MAD. These authors focused on the total extent of obstruction release as a potential predictor for treatment success. According to our findings, for the M recommendation, it is essential to identify the most similar expected response in every VOTE level to be as precise as possible for making the recommendation.

Similarly, a dental impression with the maximum comfortable protrusion for the patient has been recommended as an advancement maneuver during sedation [34,35,53]. In addition to holding the mandible to prevent its dislodgement during the entire test, several simulation bites must be removed from the mouth and reinserted to assess changes when introducing different advancements. Performing these manipulations could cause stimulation of the patient that may produce arousal. Furthermore, a remotely controlled mandibular positioner has been used during sedation to determine the effective target protrusive position within 45 min [54]. The authors found some limitations using this device: for some patients, it failed to register the full range of mandibular movement, the trays suffered slippage due to the forces exerted by the mandible losing the anchorage, and slow continuous protrusion was not possible. Possibly, these limitations were present because this device is not intended for use in sedated patients. 

By contrast, with the SAM device, the interexaminer bias of the jaw thrust maneuver is eliminated. The jaw can be moved manually forward or backward as slow (to avoid potential arousal) or as fast as needed from zero to maximum tolerable protrusion without being removed from the patient’s mouth. Although clenching occurred in some patients, the designed manually controlled mechanism of traction by which the mandible is advanced was effective in performing the protrusive movement in all the patients. All the examinations could be performed regardless of the range of anteroposterior movement as its amplitude ranged between –20 to +20 mm, which is sufficient to adapt to any anatomy and mandibular mobility. While UA behavior is analyzed, the device maintains a fixed chosen mandibular position in supine or lateral decubitus position. The upper and lower trays can be removed independently if needed or in an emergency to leave the airway free. Nevertheless, DISE was uneventful in our patients with no need for anticipated SAM extraction. In addition, with the SAM, DISE can be performed in 15–20 min, thus reducing examination time.

Following our protocol, the SAM reproduces the minimum vertical dimension needed to use a MAD. For some authors, patient’s preference was clearly in favor of the lower vertical dimension wearing a MAD [55] as a higher dimension favors jaw discomfort and may increase collapsibility of the UA [56]. During our protocol, we did have in mind that increasing the sheer size also tends to a posterior rotation causing a 0.3 mm reduction in the range of mandibular advancement for every 1 mm of vertical opening [57]. Therefore, the mandibular protrusion capacity should not be limited as it is essential for successful treatment for most patients [13,58].

Despite the fact that some authors [59,60,61] conclude that functional endoscopy is an optimal predictor for MAD treatment outcomes, it could be controversial since it investigates the dynamic behavior of the upper airway during wakefulness. In addition, Okuno et al. [59,60] only focused on cross-sectional area changes at the velopharynx to make the recommendation. According to some publications [62,63,64], DISE has several advantages including safety, ease of use, and reliability, which outweigh awake exploration to diagnose sites of obstruction and the pattern of UA collapse. It also has strong informative evidence on changes in the upper airway when introducing mandibular advancement maneuvers.

This study is no exception in having limitations. First, this work investigates the effect of MAD on DISE not during natural sleep. By using propofol, we could not reproduce REM sleep. In addition, a follow-up study of patients treated with MAD after this DISE-SAM protocol is required to verify the success of the recommendations. In future research agenda, it is mandatory to confirm with control sleep studies the utility of the DISE-SAM protocol to select patients for MAD treatment and verify the usefulness of this procedure in the selection of combination of therapies. 

## 6. Conclusions

The SAM device allowed a reproducible, measurable, and titratable maneuver to maintain a stable position during DISE. Given that there was no relationship between OSA severity and SAM recommendation and there is a weak relationship between severity and the required advancement, therefore, a precise mandibular maneuver during DISE is needed to accurately identify the potential responders.

Using the DISE-SAM protocol, the response to mandibular advancement under sedation regardless of severity could be identified. The protocol also allows measuring the range of mandibular protrusion in which improvement of UA collapsibility could be achieved. 

Therefore, the selection of therapeutic strategies should be a highly individualized process. According to SAM recommendation, MAD could be proposed within the therapeutic armamentarium for sleep-disordered breathing in the most personalized design as a single, alternative, or multiple treatment approaches.

## Figures and Tables

**Figure 1 jcm-11-00658-f001:**
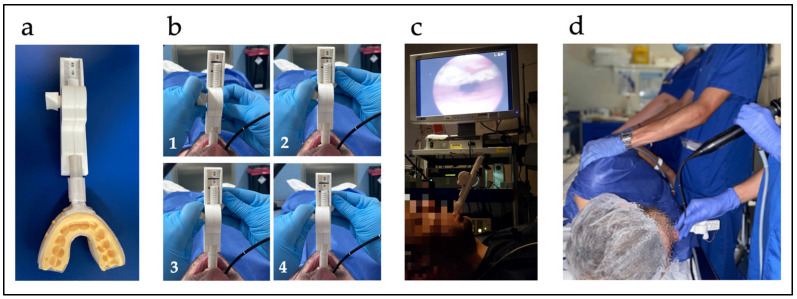
(**a**) SAM device. (**b**) SAM titration (1–4): 0%, 50%, 75%, and 100% advancement, respectively. (**c**) Patient examination in supine position. (**d**) Patient examination in lateral decubitus position.

**Figure 2 jcm-11-00658-f002:**
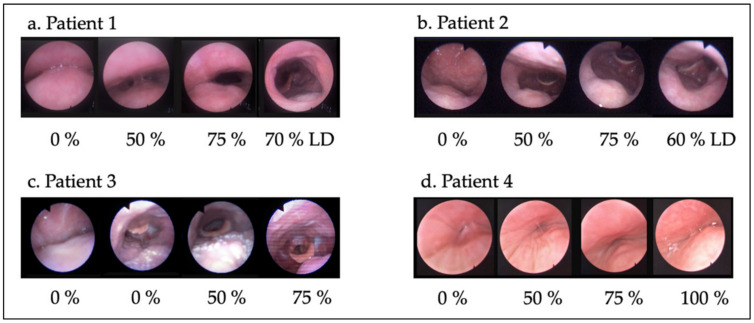
UA responses to different mandibular advancements and changes in body position. UA in lateral decubitus position was analyzed with the minimum effective advancement. (**a**) Patient 1: the combination of MAD + positional therapy significantly improved collapsibility. (**b**) Patient 2: mandibular advancement was very effective in supine position at the velopharynx, but in right lateral decubitus, an epiglottis collapse appears. (**c**) Patient 3: multilevel collapse successfully resolved with the SAM in supine position. (**d**) Patient 4: mandibular advancement was not effective.

**Figure 3 jcm-11-00658-f003:**
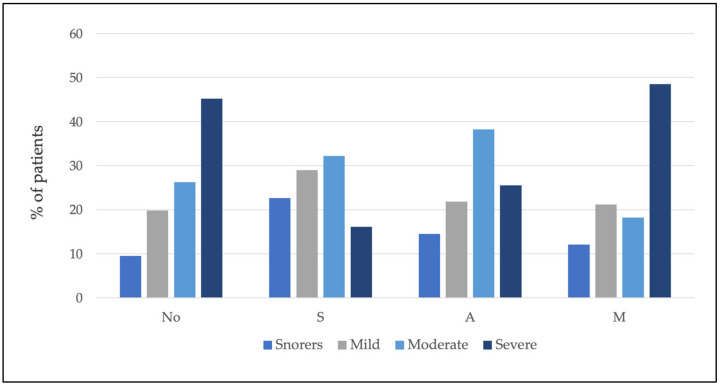
Within each level of recommendation, the percentage of patients according to severity is shown. Note that the total for each recommendation is 100%.

**Figure 4 jcm-11-00658-f004:**
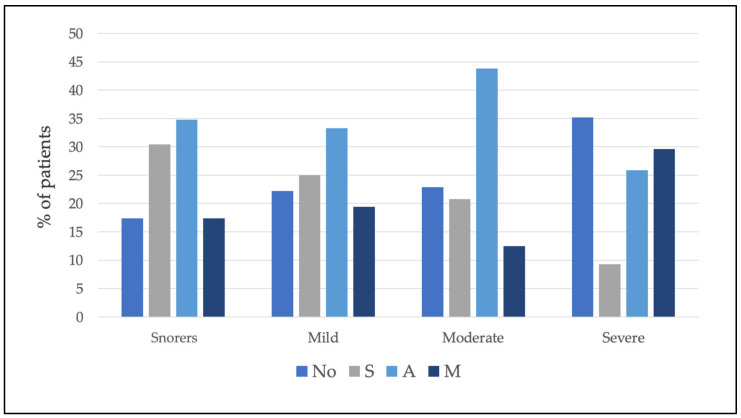
Within each severity group, the percentage of patients classified by recommendation is shown. Note that the total for each severity level is 100%.

**Figure 5 jcm-11-00658-f005:**
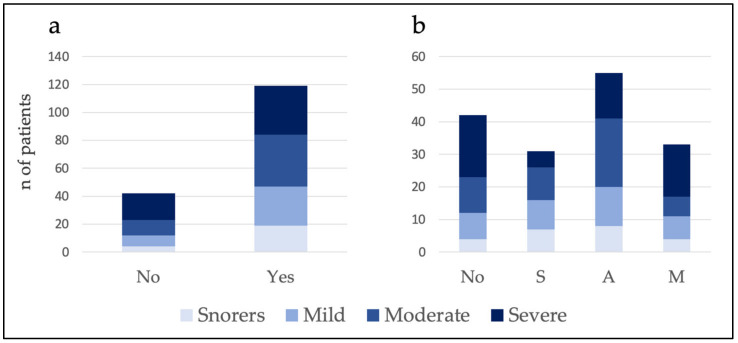
Histograms showing the number of patients for whom the MAD YES/NO recommendation (**a**) and SAM recommendation (**b**) were made. No: not recommended; S: single use; A: alternative treatment; M: multiple treatment approach.

**Figure 6 jcm-11-00658-f006:**
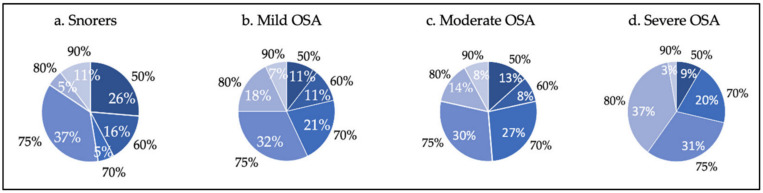
Percentages of advancement (in black) to obtain response to mandibular advancement, and percentage of patients (in white) who responded to each percentage of advancement in each severity group (**a**) Snorers; (**b**) Mild OSA; (**c**) Moderate OSA, and (**d**) Severe OSA).

**Table 1 jcm-11-00658-t001:** SAM recommendations (single use—alternative treatment—multiple treatment approach). * Greater predictive value in young patients, BMI < 30, without predominance of apnea in REM sleep and with predominance of obstructive hypopnea vs. apnea in the sleep study.

S Recommendation (single use)	As a single treatment if collapses are eliminated for patients with simple snoring or OSA. *
A Recommendation (alternative treatment)	As an alternative treatment if the collapses are improved by at least 50%, from 2 to 1 or from 1 to 0, with some VOTE levels remaining at 1. Combination of therapies may improve results. Partial response expected. *
M Recommendation (multiple treatment approach)	ONLY as a combined treatment. *
No Recommendation (MAD not recommended)	MAD is not recommended if no change or worsening is visualized.

**Table 2 jcm-11-00658-t002:** Statistical properties of the four severity groups.

	Snoring Group	Mild	Moderate	Severe
*n*	23	36	48	54
Gender (m/f)	20/3	25/11	42/6	45/9
Mean age (SD)	42.74 (9.65)	46.39 (13.12)	46.27 (11.72)	49.42 (10.22)
Mean BMI (SD)	26.06 (3.03)	26.7 (3.22)	27.82 (3.87)	29.96 (4.98)
Mean AHI (SD)	2.49 (1.80)	9.01 (2.61)	23.28 (4.41)	51.28 (15.73)
Mean ESS (SD)	8.33 (3.52)	9.52 (5.49)	9.21 (5.45)	11.52 (6.09)

AHI: apnea–hypopnea index; BMI: body mass index; ESS: Epworth sleepiness scale, M: mean; SD: standard deviation.

**Table 3 jcm-11-00658-t003:** Proportion and number of patients classified by severity within each level of recommendation.

	Snorers % (*n*)	Mild % (*n*)	Moderate % (*n*)	Severe % (*n*)	Total % (*n*)
S recommendation	22.6 (7)	29.0 (9)	32.2 (10)	16.1 (5)	100 (31)
A recommendation	14.5 (8)	21.8 (12)	38.2 (21)	25.5 (14)	100 (55)
M recommendation	12.1 (4)	21.2 (7)	18.2 (6)	48.5 (16)	100 (33)
No recommendation	9.5 (4)	19.0 (8)	26.2 (11)	45.2 (19)	100 (42)

**Table 4 jcm-11-00658-t004:** Proportion and number of patients classified by recommendation within each level of severity.

	Snorers % (*n*)	Mild % (*n*)	Moderate % (*n*)	Severe % (*n*)	Total % (*n*)
S recommendation	30.4 (7)	25 (9)	20.8 (10)	9.3 (5)	19.3 (31)
A recommendation	34.8 (8)	33.3 (12)	43.8 (21)	25.9 (14)	34.2 (55)
M recommendation	17.4 (4)	19.4 (7)	12.5 (6)	29.6 (16)	20.5 (33)
No recommendation	17.4 (4)	22.2 (8)	22.9 (11)	35.2 (19)	26.1 (42)
					100 (161)

## Data Availability

The data are contained within the article.

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
