# Peer review of "Optimizing Mandibular Advancement Maneuvers during Sleep Endoscopy with a Titratable Positioner: DISE-SAM Protocol"

_jcm, 2022, doi:10.3390/jcm11030658_

Round 1

Reviewer 1 Report

The use of a titratable SAM device during DISE is of interest for the field of dental sleep medicine. This article describes the protocol and methodology of using this titratable, reproducible, and measurable maneuver during DISE. It aimed to find the relationship between the severity of OSA and the degree of response with SAM, and to determine the advancement required to improve the collapsibility of the upper airway.

Although the methodology is of interest for the field of dental sleep medicine, my main question is the link between the treatment recommendation after the DISE and the real treatment outcome of MAD (with or without positional therapy). Perhaps it was not the main aim of the study, but this link is missing to assess the exact value of the SAM procedure in predicting the best protrusive position or setting the treatment recommendation.

  • What was the effect of MAD in the titrated position during DISE?
  • What was the effect of the recommended therapy on the respiratory disease?

 Please comment on this, and add this to the discussion section.

The study included 161 with snoring and/or OSA:

  • Was a power analysis performed?
  • In how many patients the SAM – DISE procedure was not successful? Did some patients experienced clenching, making it hard to perform the titration manually during DISE?
  • Why include patients with non-apneic or primary snoring? It is hard to see apneas during DISE in those patients as you expect less than 5 apneas / hour !

What is the time frame of the DISE procedure in most patients, including all these manoeuvres?

All patients also underwent a baseline DISE procedure: was there a correlation between the baseline findings during DISE and the recommendation after SAM procedure? In other words; is there a specific type of collapse or combination of collapses (multilevel collapse) that was harder to overcome with the SAM procedure?

The protocol is confusing: it is stated that advancement positions of 0%, 50%, 75% and 100% were consecutively evaluated. However, in figure 2, it is stated that some patients are only in 0%-50% range and than 70%. Please rewrite the procedure.

Table 1 is confusing. Although the classification is interesting; consider re-writing the table text to make it more simple.

In the discussion: assumptions are made on complete or partial response of MAD therapy in those patients, however, there is no follow-up sleep study performed so please be cautious with these assumptions

Line 196: Please replace IAH by AHI

Line 216: Please replace EES by ESS

Reviewer 2 Report

Thank you for this interpretation of the DISE methodology.

Only 3 considerations

1) chapter materials and methods

 I would insert more details regarding the characteristics and management of the Selector Avance Mandibular device (SAM), i.g. materials, manufacture,  disposable...

  2) chapter limitations

Considering that the level of sedation is influenced by the positioning of the SAM and by changes in the patient's decubitus, I would insert the difficulty/limitation of maintaining homogeneous (as well as correct) BIS values during the diagnostic phases of the DISE procedure.

The use of SAM certainly reduces interobserver bias but probably the complete solution to this problem would be found in the simultaneous recording of BIS values during the DISE-SAM protocol.

3) the endoscopic image of the patient 4 - 60%, lateral decubitus (line 173) - should be improved.

Round 2

Reviewer 1 Report

I would like to thank the authors for their extensive reply to all the comments and suggestions. In my opinion, the clarity of the paper improved sufficiently.

I do have only one additional comment:

I do understand that the follow-up sleep study is work in progress or work for future studies. This is stated as limitation of the study in the discussion part. However, I would suggest to change the title accordingly, since it is to strong to state 'prediction tool of treatment outcome' since you don't measure treatment outcome itself.

Author Response

We thank the reviewers for their careful reading of our manuscript and their many insightful comments and suggestions. Below we respond to the comments of each reviewer in detail. We are also providing a revised manuscript that reflects their suggestions and comments. We feel that this has strengthened the manuscript.

Revisor #1

Q0.-I would like to thank the authors for their extensive reply to all the comments and suggestions. In my opinion, the clarity of the paper improved sufficiently.

I do have only one additional comment:

I do understand that the follow-up sleep study is work in progress or work for future studies. This is stated as limitation of the study in the discussion part. However, I would suggest to change the title accordingly, since it is to strong to state 'prediction tool of treatment outcome' since you don't measure treatment outcome itself.

R0.- Following this suggestion, we have changed the title into the following:

Optimizing mandibular advancement maneuvers during sleep endoscopy with a titratable positioner: DISE-SAM protocol.